# Chirality, magic, and quantum correlations in multipartite quantum states

Shreya Vardhan,[1, *] Bowen Shi,[2, 3, 4] Isaac H. Kim,[4] and Yijian Zou[5, †]

[1]*Stanford Institute for Theoretic Physics, Stanford University, Stanford, CA 94305, USA*
[2]*Department of Physics, University of Illinois, Urbana, Illinois 61801, USA*
[3]*Department of Physics, University of California at San Diego, La Jolla, CA 92093, USA*
[4]*Department of Computer Science, University of California, Davis, CA 95616, USA*
[5]*Perimeter Institute for Theoretical Physics, Waterloo, Ontario N2L 2Y5, Canada*
(Dated: December 1, 2025)

Chirality is a fundamental property of many topological phases, yet it lacks a general information-theoretic formulation. In this work, we introduce a notion of chirality for generic quantum states, defined by the impossibility of transforming a state into its complex conjugate under local unitary operations. We propose several quantitative measures of chirality, including a faithful metric called the chiral log-distance, and a family of nested commutators of modular Hamiltonians. We show that chirality, although not a resource in the traditional sense, is intrinsically linked to two major classes of quantum resources: magic and quantum correlations. In particular, we demonstrate that (i) qubit stabilizer states are always non-chiral, (ii) the chiral log-distance provides a lower bound for several magic monotones, and (iii) a nested commutator-based chirality measure is lower bounded by a variant of interferometric power.

## I. INTRODUCTION

Chirality refers to the lack of invariance between certain physical systems and their mirror images. This property naturally appears in a variety of quantum many-body systems and quantum field theories, and underlies a wide range of interesting physical phenomena. For example, in condensed matter physics, topological insulators and topological superconductors [1] have chiral edge states which transport charge only in one direction. In high-energy physics, chiral fermions are essential to electroweak interactions in the Standard Model [2]. In many-body systems, chirality is identified from macroscopic properties, such as the direction of the energy current or charge current. From a microscopic perspective, it is natural to ask how chirality can be characterized in terms of the structure of the quantum state.

In this paper, we introduce a natural information-theoretic definition of chirality as an intrinsic property of a general quantum state. Many-body states such as the edge states of quantum (thermal) Hall insulators and superconductors [3–5], which are "macroscopically" chiral due to the direction of the charge or energy current, are also chiral by our microscopic definition. [1] While these many-body settings provide the motivation for the definition, in this paper we will explore chirality as a general property of quantum states, including those of few-body systems. Our goal is to

understand whether and how this notion of chirality is related to other features of the quantum state, such as its entanglement, nonstabilizerness, and discord-like quantum correlations.

Intuitively, a chiral state is one that is sharply distinguishable from its time-reversal. Recall that time-reversal is an anti-unitary operation, which is often defined to be complex conjugation with respect to a basis which depends on the physical context. To give an intrinsic *basis-independent* definition of chirality for a quantum state, we will find it useful to define some partition of the Hilbert space into $n$ subsystems, $A_1, ..., A_n$. With respect to this partition, we propose the following definition:

**Definition 1** (Chiral quantum state)**.** Consider an $n$-partite state $\rho_{A_1 A_2 \cdots A_n}$, and some arbitrary product basis $|s_1\rangle_{A_1} \otimes ... \otimes |s_n\rangle_{A_n}$ between the subsystems. Consider the complex conjugation $\rho^*_{A_1...A_n}$ of $\rho_{A_1...A_n}$ in this basis.

$\rho_{A_1 A_2 \cdots A_n}$ is *n-partite chiral* if

$$\rho^*_{A_1 A_2 \cdots A_n} \neq (\otimes_{i=1}^n U_{A_i}) \, \rho_{A_1 A_2 \cdots A_n} \, (\otimes_{i=1}^n U_{A_i}^\dagger) \quad (1)$$

for any set of local unitary operators $\{U_{A_i}\}$.

Note that the definition of chirality is independent of the particular product basis $|s_1\rangle_{A_1} \otimes ... \otimes |s_n\rangle_{A_n}$ chosen for the complex conjugation as we allow all possible local unitaries in (1).

We first introduce a variety of measures to detect and quantify chirality in a given state. Starting with the above definition, a natural way to quantify chirality is using the minimum distance between the LHS and RHS of (1) for all possible choices of the $\{U_{A_i}\}$.

---

* vardhan@stanford.edu

† yzou@perimeterinstitute.ca

[1] This statement can be seen from the results of the previous works [6–9], as we will discuss more explicitly later.

We introduce a measure called the *chiral log-distance* (7) based on this idea. While this measure is intuitive and useful for certain proofs, it has the drawback that it requires an optimization over the set of all local unitaries, and does not have an explicit computable expression in terms of the density matrix. To address this issue, we further introduce a family of more easily computable measures called *nested commutator measures*, which involve commutators between modular Hamiltonians associated with reduced density matrices of various subsystems. These measures further have the property that they are additive under tensor product. The "modular commutator" introduced and studied in [6–10] in various quantum many-body systems can be seen as an example of such a measure for $n = 3$, which in particular shows that edge states of quantum Hall insulators and superconductors are tripartite chiral. Another computable quantity previously introduced in the literature, which can detect tripartite chirality, is the rotated Petz map fidelity. This measure concerns the fidelity of a particular recovery operation for the erasure of $C$ in the tripartite state $\rho_{ABC}$ [11, 12]. It has been demonstrated that the fidelity is asymmetric with respect to the rotation parameter for chiral edge states, but symmetric for non-chiral states [13, 14].

On an intuitive level, chirality is an inherently quantum property of the state due to the crucial role played by complex numbers in the wavefunction in Definition 1. A natural question is whether chirality can be seen as a quantum resource [15] in a manner similar to resource-theoretic characterizations of entanglement [16] or non-stablizerness [17]. A resource-theoretic formulation involves a setup with certain free states and free operations, and defines "resource states" as those that cannot be prepared from free states and operations. For example, in the resource theory of entanglement, separable states are free states, local operations and classical communication (LOCC) are free operations, and entangled states are resource states.

We provide evidence that it may not be possible to fit chirality within the framework of standard resource theories by taking non-chiral states to be free states and chiral states to be resource states. We will show that $n-$partite non-chiral states can be mapped to chiral states with the same partition by partial traces within subsystems, which should intuitively be included in the set of free operations. We will further show that while chirality in pure states necessarily requires quantum entanglement, as can immediately be seen from Definition 1, chirality in mixed states does not require quantum entanglement. Indeed, we will provide explicit examples of separable bipartite states which are chiral. On the other hand, we will show

that measures of chirality provide lower bounds on two other kinds of quantum resources in general quantum states: *magic* and *discord-like quantum correlations*.

*Magic* refers to the amount of "non-stabilizer resource" present in a quantum state. Stabilizer states are a subset of all quantum states which can be efficiently simulated on a classical computer, despite in general being highly entangled [18]. Non-stabilizer states are therefore a fundamental resource for universal quantum computation, and the usefulness of a given state for this purpose can be rigorously quantified by measures called "magic monotones" [17, 19], which are non-increasing under free operations known as stabilizer protocols. A number of interesting quantities have been used to quantify magic(see for instance [17, 19–25]. In particular, in this paper, we will make use of the stabilizer nullity [23], the min-relative entropy of magic [21], and the basis-minimized stabilizer asymmetry (BMSA) [24]. [2]

In this paper, we show that if a quantum state is chiral, then it necessarily contains magic or nonstabilizer resource, and moreover the extent of chirality provides a lower bound on the amount of non-stabilizerness. More explicitly, we show that (i) all stabilizer states of qubits, pure or mixed, are non-chiral, and (ii) for pure states, the chiral log-distance provides a lower bound on the BMSA, which can in turn lower-bound various other magic monotones, including the stabilizer nullity and min-relative entropy of magic.

*Discord-like quantum correlations* provide a general notion of quantum correlations beyond entanglement [26]. Such correlations may be present in certain separable mixed states. In a resource theory formulation for such quantum correlations [27], the free states are a subset of separable states known as "classical-quantum states." For a given bipartition of a system, a classical-quantum state is defined as a state for which there exists a complete projective measurement within one of the subsystems under which the state is invariant. A number of quantities have been introduced to quantify discord-like correlations. We introduce a new measure of such correlations called the *intrinsic interferometric power*, closely related to the interferometric power of [28, 29]. Unlike the interferometric power, this quantity is defined purely in terms of properties of the state and without reference to an external Hamiltonian or energy spectrum. We show that our nested commutator measures for chirality can be used to provide a lower bound on the intrinsic interferometric power. We summarize the relation between bipar-

―――――

[2] The stabilizer nullity and min-relative entropy of magic are magic monotones by the definition of [17]. The BMSA is a magic monotone by the definition recently introduced in [19].

tite chiral states and known classes of bipartite states in Fig. 1. [3]

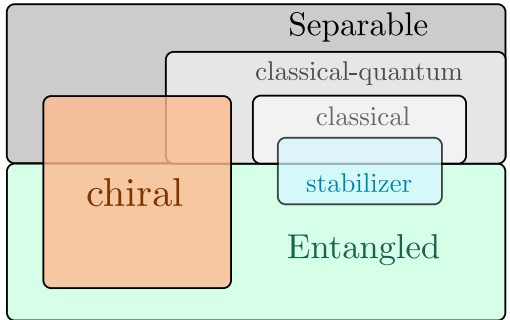

FIG. 1. Bipartite chiral states can either be separable or entangled. Classical states (Eq. (61)) are nonchiral, but classical-quantum states (Eq. (60)) may be chiral. Chiral states of qubits are always non-stabilizer.

The plan of the paper is as follows. In Section II, we discuss some examples of few-body chiral states and introduce various measures of chirality, including the chiral log-distance and nested commutator measures. We also discuss some general consequences of Definition 1, including the non-monotonicity of chirality under local partial traces. In Section III, we study the relation between chirality and magic. In Section IV, we study the relation between chirality and discord-like quantum correlations. In Section V, we conclude with possible applications of our results to quantum many-body dynamics and quantum phases of matter.

## II. EXAMPLES, GENERAL PROPERTIES, AND MEASURES OF CHIRALITY

In this section, we develop a general framework for studying chirality as an information-theoretic property of multipartite quantum states. In Sec. II A, we construct an explicit example of a chiral few-body state, and use it to illustrate certain general features of the relation between $n$-partite chirality for different $n$. In Sec. II B, we discuss general requirements for a measure of chirality, and introduce a measure which we call the chiral log-distance, which is physically intuitive but not easily computable. In Sec. II C, we introduce computable and additive measures of chirality constructed from nested commutators, which are odd under complex conjugation of the state in a local product basis. In Sec. II D, as an illustration, we evaluate these computable measures in random two-qubit mixed states,

———

[3] Stabilizer states do not have intersections with separable states that is beyond classical states, as noted in Ref. [30].

and in particular note the lack of correlation between chirality and entanglement.

### A. Explicit examples and general properties of chirality

From Definition 1, any state $\rho$ is non-chiral for $n = 1$. Similarly, $n$-partite product states are always $n$-partite non-chiral. Further, it can be seen using the Schmidt decomposition that any pure state is non-chiral for $n = 2$. The simplest non-trivial case where we can look for examples of chiral states is therefore that of a bipartite mixed state. Let us first provide an explicit example which can be checked to be chiral directly from the definition:

**Example 1.** *The following state of a qutrit $A$ and a qubit $B$ is bipartite chiral:*

$$\rho_{AB} = \sum_{i=1}^{3} p_i |i\rangle\langle i|_A \otimes |\psi_i\rangle\langle\psi_i|_B \qquad (2)$$

*where $\{|i\rangle_A\}$ is an orthonormal basis for $A$, the $p_i$'s are nondegenerate, $p_i > 0$, and*

$$|\psi_1\rangle = |0\rangle, |\psi_2\rangle = \frac{1}{\sqrt{2}}(|0\rangle + |1\rangle), |\psi_3\rangle = \frac{1}{2}(|0\rangle + \sqrt{3}i|1\rangle). \qquad (3)$$

To see that (2) is chiral, assume for contradiction that one can find $U_A, U_B$ such that $(U_A \otimes U_B)\rho_{AB}(U_A^\dagger \otimes U_B^\dagger) = \rho_{AB}^*$. We can check that since the $p_i$'s are nondegenerate and $\rho_A^* = \rho_A$, we must have $U_A |i\rangle\langle i| U_A^\dagger = |i\rangle\langle i|$. This further implies we must have $U_B |\psi_i\rangle = e^{-i\theta_i} |\psi_i^*\rangle$. The latter implies that $\langle\psi_i|\psi_j\rangle = e^{i(\theta_i - \theta_j)}\langle\psi_i^*|\psi_j^*\rangle = e^{i(\theta_i - \theta_j)}\langle\psi_i|\psi_j\rangle^*$, which implies that $\arg\langle\psi_i|\psi_j\rangle + \arg\langle\psi_j|\psi_k\rangle = \arg\langle\psi_i|\psi_k\rangle$. However, this is not true since $\langle\psi_1|\psi_2\rangle$ and $\langle\psi_1|\psi_3\rangle$ are real while $\langle\psi_2|\psi_3\rangle$ is complex.

It is also useful to consider a purification of (2) by adding a qutrit $A'$:

$$|\psi_{AA'B}\rangle = \sum_{i=1}^{3} \sqrt{p_i} |i\rangle_A \otimes |i\rangle_{A'} \otimes |\psi_i\rangle_B \qquad (4)$$

This example provides a useful illustration of two general features of chirality. Note that:

1. $|\psi_{AA'B}\rangle$ is a *tripartite chiral state*. This is an example of the more general fact in Proposition 1 below.

2. For a bipartition of the system into $AA'$ and $B$, the state (4), like any pure state, is bipartite non-chiral. This immediately shows that bipartite chirality does not monotonically decrease

under the action of local partial traces. By applying a partial trace in $A'$, (4) can be mapped to the bipartite chiral state (2). It is straightforward to generalize this example to show that $n$-partite chirality does not monotonically decrease under local partial traces. Nevertheless, we will show that chirality satisfies a notion of monotonicity when we allow the number of partitions to change, see Proposition 2 below.

**Proposition 1.** *A pure state $|\psi_{A_1 A_2 \cdots A_n}\rangle$ is $n$-partite chiral, if and only if its $(n-1)$-partite reduced density matrix $\rho_{A_1 A_2 \cdots A_{n-1}}$ is $(n-1)$-partite chiral.*

*Proof.* We only need to prove the contrapositive statement: a pure state $|\psi_{A_1 A_2 \cdots A_n}\rangle$ is $n$-partite non-chiral, if and only if its $(n-1)$-partite reduced density matrix $\rho_{A_1 A_2 \cdots A_{n-1}}$ is $(n-1)$-partite non-chiral. The proof of the "only if" part goes as follows. Suppose $|\psi_{A_1 A_2 \cdots A_n}\rangle$ is non-chiral, then there exists a set of unitaries $\{U_{A_i}\}_{i=1}^n$ whose tensor product maps $|\psi\rangle$ to $|\psi^*\rangle$. It follows immediately that conjugation by $U_{A_1} \otimes \cdots \otimes U_{A_{n-1}}$ takes $\rho_{A_1 A_2 \cdots A_{n-1}}$ to $\rho^*_{A_1 A_2 \cdots A_{n-1}}$. The proof of the "if" part goes as follows. Let the tensor product of $\{V_{A_i}\}_{i=1}^{n-1}$ be the unitary that maps $\rho_{A_1 A_2 \cdots A_{n-1}}$ to $\rho^*_{A_1 A_2 \cdots A_{n-1}}$ by conjugation. Then the state $|\varphi\rangle := V_{A_1} \otimes \cdots \otimes V_{A_{n-1}}|\psi\rangle$ must reduce to $\rho^*_{A_1 A_2 \cdots A_{n-1}}$, and this is identical to the reduced density matrix of $|\psi^*\rangle$ on $A_1 A_2 \cdots A_{n-1}$. Then, by Uhlmann's theorem [31], there exists a unitary $W_{A_n}$ such that $|\psi^*\rangle = W_{A_n}|\varphi\rangle$. Thus, $|\psi_{A_1, A_2, \cdots, A_n}\rangle$ is $n$-partite non-chiral. This completes the proof. $\square$

If we know that a state is $(n-1)$-partite chiral, then we know it is also $n$-partite chiral on dividing any of the $n-1$ subsystems further into two. In this sense, bipartite chirality is the strongest form of chirality for mixed states, and tripartite chirality is the strongest form of chirality for pure states. We can define a mixed (pure) state as being chiral independent of partition if it is chiral for all possible bipartitions (tripartitions).

### B. Chirality measures

For a general quantum state, a simple analysis such as that of Example 1 is not possible, and it is nontrivial to determine whether or not it is chiral. To address this, it is useful to introduce quantities whose nonvanishing can detect chirality. Moreover, it would be useful to quantify the extent to which a given state is chiral. In this section, we introduce a few different measures to detect and quantify chirality.

We define an $n$-partite chirality measure as a map from $\rho$ to a real number, $\mathcal{A}_{A_1 A_2 \cdots A_n}(\rho) \in \mathbb{R}$, satisfying two properties:

1. It is invariant under local unitaries:

$$\mathcal{A}_{A_1 A_2 \cdots A_n}(\rho) = \mathcal{A}_{A_1 A_2 \cdots A_n}(\rho') \quad \text{for } \rho \sim \rho' \qquad (5)$$

where $\rho \sim \rho'$ refers to $\rho' = (\otimes_i U_{A_i}) \rho (\otimes_i U_{A_i}^\dagger)$ for some unitary operators $\{U_{A_i}\}_{i=1}^n$.

2. It vanishes for non-chiral states:

$$\mathcal{A}_{A_1 A_2 \cdots A_n}(\rho) = 0 \quad \text{for } \rho \sim \rho^*. \qquad (6)$$

We can consider chirality measures that are even under $\rho \to \rho^*$, as well as those that are odd under $\rho \to \rho^*$.[4] Below, we consider the case of bipartite chirality in mixed states. For a tripartite pure state $|\psi_{ABC}\rangle$, these bipartite measures for the reduced density matrix $\rho_{AB}$ can also be used to quantify tripartite chirality of $|\psi_{ABC}\rangle$. The generalization to $n$-partite chirality is straightforward.

One natural choice of chirality measure is the following quantity, which we will call *the chiral log-distance*:

$$C_{A|B}(\rho_{AB}) = -\log \max_{U_A, U_B} F(\rho^*_{AB}, U_A \otimes U_B \rho_{AB} U_A^\dagger \otimes U_B^\dagger),$$
$$(7)$$

where the maximization is over unitary operators $U_A$, $U_B$, and $F(\rho, \sigma) := \|\sqrt{\rho}\sqrt{\sigma}\|_1^2$ is the Uhlmann fidelity. The log-distance can be motivated directly from the definition of chiral states. The quantity is non-negative, and it is zero if and only if $\rho$ is nonchiral, i.e., the measure is faithful. It is even under $\rho \leftrightarrow \rho^*$, that is,

$$C_{A|B}(\rho^*) = C_{A|B}(\rho). \qquad (8)$$

We note that $C_{A|B}(\rho)$ can be recast as a solution to the following optimization problem using Uhlmann's theorem [31]:

$$\exp(-C_{A|B}(\rho_{AB}))$$
$$= \max_{U_A, U_B, U_C} |\langle\psi_{ABC}|(U_A \otimes U_B \otimes U_C)|\psi^*_{ABC}\rangle|, \qquad (9)$$

where $|\psi\rangle$ and $|\psi^*\rangle$ are purifications of $\rho$ and $\rho^*$, respectively. We denote the purifying system as $C$. Interestingly, the optimization over individual $U_A, U_B$, and $U_C$ can be solved exactly. For instance, suppose we fix $U_B$ and $U_C$. Then the objective function can be written as the absolute value of $\mathrm{Tr}(U_A O_A)$ for some matrix $O_A$. We can simply take the singular value decomposition of $O_A = U\Sigma V^\dagger$ and choose $U_A = VU^\dagger$.

---

[4] Note that any chirality measure can be written as the sum of an even part and an odd part: $\mathcal{A} = C + J$, where $C$ stands for the even part and $J$ stands for the odd part. That is, $C(\rho) = (\mathcal{A}(\rho) + \mathcal{A}(\rho^*))/2$ and $J(\rho) = (\mathcal{A}(\rho) - \mathcal{A}(\rho^*))/2$.

However, Eq. (9) is nonetheless not a global convex optimization problem. As such, solving it requires using a gradient-based algorithm or a greedy algorithm. These algorithms may fall into local extrema, in which case only a strict lower bound of $C_{A|B}$ can be obtained.

Another nice property of the chiral log-distance is that it is monotonic under tracing out an entire party *together with discarding the party in the partition*. This can be formalized in the following proposition.

**Proposition 2.** *(Monotonicity of chiral log-distance under partial trace)*

$$C_{A_1 A_2 \cdots A_n}(\rho_{A_1 A_2 \cdots A_n}) \geq C_{A_2 \cdots A_n}(\rho_{A_2 \cdots A_n}) \tag{10}$$

*Proof.* By definition,

$$C_{A_1 A_2 \cdots A_n}(\rho_{A_1 A_2 \cdots A_n}) = -\log \max_{\{U_{A_i}\}} F((\otimes_{i=1}^N U_{A_i})\rho_{A_1 A_2 \cdots A_n}(\otimes_{i=1}^N U_{A_i}^\dagger), \rho_{A_1 A_2 \cdots A_n}^*) \tag{11}$$

Suppose $U_{A_1} \cdots U_{A_n}$ achieves the maximum above, then

$$
\begin{aligned}
C_{A_1 A_2 \cdots A_n}(\rho_{A_1 A_2 A_3 \cdots A_n}) &= -\log F((\otimes_{i=1}^n U_{A_i})\rho_{A_1 A_2 \cdots A_n}(\otimes_{i=1}^n U_{A_i}^\dagger), \rho_{A_1 A_2 \cdots A_n}^*) \\
&\geq -\log F((\mathrm{Tr}_{A_1}(\otimes_{i=1}^n U_{A_i})\rho_{A_1 A_2 \cdots A_n}(\otimes_{i=1}^n U_{A_i}^\dagger)), \mathrm{Tr}_{A_1}\rho_{A_1 A_2 \cdots A_n}^*) \\
&= -\log F((\otimes_{i=2}^n U_{A_i})\rho_{A_2 \cdots A_n}(\otimes_{i=2}^n U_{A_i}^\dagger), \rho_{A_2 \cdots A_n}^*) \\
&\geq -\log \max_{\{V_{A_i}\}} F((\otimes_{i=2}^n V_{A_i})\rho_{A_2 \cdots A_n}(\otimes_{i=2}^n V_{A_i}^\dagger), \rho_{A_2 \cdots A_n}^*) \\
&= C_{A_2 \cdots A_n}(\rho_{A_2 \cdots A_n})
\end{aligned}
\tag{12}
$$

In the second line we use the fact that fidelity is monotonically increasing under partial trace and in the fourth line we take the maximum over all local unitaries for the $(n-1)-$partite state. $\square$

Note that this does not indicate that $n$-partite chirality is monotonic under a local channel of one party, as we noted in Sec. II A. What monotonicity does guarantee is that any purification of an $n$-partite chiral mixed state must be $(n+1)$-partite chiral, consistent with Proposition 1.

Below, we will also consider chirality measures $J : \rho \to \mathbb{R}$ which are *odd* under $\rho \to \rho^*$, i.e.,

$$J_{A|B}(\rho^*) = -J_{A|B}(\rho). \tag{13}$$

Considering odd chirality measures is natural from the perspective that they can distinguish states with "left" and "right" directionality. Such measures are aligned with the physical motivation for chirality and similar to quantities such as charge and energy currents which typically detect chirality in physical setups.[5] Moreover, we will show below that only odd measures can

satisfy the following requirement of *additivity*:

$$\mathcal{A}_{AA'|BB'}(\rho_{AB} \otimes \rho_{A'B'}) = \mathcal{A}_{A|B}(\rho_{AB}) + \mathcal{A}_{A'|B'}(\rho_{A'B'}). \tag{14}$$

**Lemma 1** (Additivity)**.** *A chirality measure $\mathcal{A}_{A|B}$ is additive only if it is odd under $\rho \to \rho^*$.*

*Proof.* Suppose a chirality measure $\mathcal{A}_{A|B}$ is additive. Let $A'$ and $B'$ to be identical copies of $A$ and $B$.

$$\mathcal{A}_{AA'|BB'}(\rho_{AB} \otimes \rho_{A'B'}^*) = \mathcal{A}_{A|B}(\rho) + \mathcal{A}_{A'|B'}(\rho^*) \tag{15}$$

by additivity. On the other hand, we can construct local SWAP (unitary) gates $\mathrm{SWAP}(A, A')$ and $\mathrm{SWAP}(B, B')$ such that $\rho_{AB} \otimes \rho_{A'B'}^*$ is converted to $\rho_{AB}^* \otimes \rho_{A'B'}$, which implies that $\mathcal{A}_{AA'|BB'}(\rho_{AB} \otimes \rho_{A'B'}^*) = 0$. Plugging this back in Eq. (15), we get $\mathcal{A}_{A|B}(\rho) = -\mathcal{A}_{A|B}(\rho^*)$. Thus, any additive chirality measure must be odd. $\square$

In the next subsection, we will discuss a general recipe for constructing odd bipartite chirality measures $J_{A|B}$, including additive measures, by using nested commutators of reduced density matrices on different

---

[5] For tripartite states, the modular commutator $J(A, B, C)_\rho := i\mathrm{Tr}([\log \rho_{AB}, \log \rho_{BC}]\rho_{ABC})$ is one such example [7].

subsystems. These measures will further have the advantage that they have explicit expressions in terms of the density matrix and therefore are computable.

### C. Measures involving nested commutators

We construct additive measures of bipartite chirality, which can be written as certain functions of the density matrices $\rho_S$ for various subsystems $S$ and their modular Hamiltonians $K_S := -\log \rho_S$. [6] Such measures are computable without optimization over local unitaries, and are odd under complex conjugation.

A previously known tripartite chirality measure, $J(A, B, C) = i\text{Tr}(\rho_{ABC}[K_{AB}, K_{BC}])$, uses a single commutator of certain modular Hamiltonians. For bipartite systems, the most naive generalization fails, as $[K_A, K_B] = 0$ and $\text{Tr}(\rho_{AB}[K_{AB}, K_A]) = \text{Tr}(\rho_{AB}[K_{AB}, K_B]) = 0$ identically. However, one may still construct additive chirality measures using multiple commutators or anticommutators involving $K_A$, $K_B$ and $K_{AB}$. One example is the following quantity:

$$J_2(\rho_{AB}) := i\text{Tr}(\rho_{AB}\{[K_{AB}, K_A], K_B\}). \tag{16}$$

One can check that $J_2$ is additive and odd under complex conjugation of $\rho_{AB}$ in a product basis. More generally, we will propose below a systematic way to construct a family of such additive chirality measures in the form

$$J_{A|B}(\rho_{AB}) = i\text{Tr}(\rho_{AB}X), \tag{17}$$

where $X$ is a real polynomial (possibly of infinite degree) involving the modular Hamiltonians $K_A$, $K_B$, $K_{AB}$.

The form (17) automatically satisfies local unitary invariance. Now let $S$ be the set of maps $X_{A|B}$ from $\rho_{AB}$ to operators on the Hilbert space which satisfy the following additivity property when $A'$, $B'$ are respectively identical copies of $A$, $B$:

$$X_{AA'|BB'}(\rho_{AB} \otimes \rho_{A'B'}) = X_{A|B}(\rho_{AB}) + X_{A'|B'}(\rho_{A'B'}). \tag{18}$$

Simple examples of such functions are $X_{A|B}(\rho_{AB}) = K_A, K_B, K_{AB}$.

Next, let us define the following subsets of $S$:

$$S_\pm = \{X_{A|B} \in S \mid X_{A|B}(\rho_{AB}^*) = \pm X_{A|B}(\rho_{AB})^T\}. \tag{19}$$

───────

[6] If $\rho$ is not full-rank, we define the log only on the support of $\rho$, i.e., $K_S = \sum_{i, \ p_i \neq 0} \log(p_i) |i\rangle \langle i|$ where $p_i$, $|i\rangle$ are eigenvalues and eigenstates of $\rho$.

We will show that additive odd chirality measures can be constructed sequentially by using nested commutators involving $K_A$, $K_B$, and $K_{AB}$. In the following, we will omit the subscript $A|B$ as the bipartition is clear from the context.

**Proposition 3** (Nested commutator measures). *The following holds for the sets $S_\pm$,*
  *(1) $K_A, K_B, K_{AB} \in S_+$*
  *(2) If $X, Y \in S_\pm$, then $\alpha X + \beta Y \in S_\pm, \forall \alpha, \beta \in \mathbb{R}$.*
  *(3) If $X \in S_a, Y \in S_b$, then $[X, Y] \in S_{-ab}$*
  *(4) If $X \in S_-$ then $J = i\,Tr(\rho_{AB}X)$ is an additive chirality measure.*
  *(5) If $X \in S_a$, $Y \in S_b$, $ab = -1$ and $Tr(\rho_{AB}X(\rho_{AB}))$ identically vanishes, then $J = i\,Tr(\rho_{AB}\{X, Y\})$ is an additive chirality measure.*

*Proof.* These statements can be directly verified using the above definitions. See appendix A for more details. □

Note that in (5), the condition $\text{Tr}(\rho_{AB}X(\rho_{AB})) \equiv 0$ can be satisfied by $X = [K_{AB}, W(K_A, K_B, K_{AB})]$, where $W$ can be any function.

Using the proposition, we can construct a number of additive chirality measures.

**Corollary 3.1.** *Let $X$ be nested commutators of $K_A, K_B$ and $K_{AB}$. If there are in total an odd number of commutators, then $J(\rho_{AB}) = i\,Tr(\rho_{AB}X)$ is an additive chirality measure. For example,*

$$J_3(\rho_{AB}) = i\,Tr(\rho_{AB}[[K_{AB}, [K_{AB}, K_A]], K_B]) \tag{20}$$

*is an additive chirality measure.*

*Proof.* This follows from (1), (3) and (4). □

**Corollary 3.2.** *$J_2$, defined in (16), is an additive chirality measure.*

*Proof.* To see this, note that $[K_{AB}, K_A] \in S_-$ follows from (1) and (3), and also $\text{Tr}(\rho_{AB}[K_{AB}, K_A]) \equiv 0$. Then, (5) ensures that $J_2$ is an additive chirality measure. □

**Remark.** Note that for all degree 2 polynomials $X \in S_-$, $\text{Tr}(\rho_{AB}X) \equiv 0$. Thus, the above chirality measure has the lowest possible degree in $X$.

Nested commutators generate chirality measures with increasing complexity. We can organize some of these measures into a single function which involves modular flow. The modular flow with respect to $\rho_{AB}$ is a one-parameter family of unitaries $U(s) = \rho_{AB}^{is}$, with $s \in \mathbb{R}$. We may generate a family of operators under modular flow as $K_A(s) = U(s)K_AU^\dagger(s)$ and $K_B(s) = U(s)K_BU^\dagger(s)$, where $s$ is a real number.

**Corollary 3.3.** $K_P^{(+)}(s) := (K_P(s) + K_P(-s))/2 \in S_+$ and $K_P^{(-)}(s) = i(K_P(s) - K_P(-s))/2 \in S_-$, where $P = A, B$. Furthermore, $Tr(\rho_{AB} K_P^{(-)}(s)) \equiv 0$ identically vanishes.

*Proof.* To see this, simply note that

$$K_P(s) = K_P + is[K_{AB}, K_P] \\ + \frac{(is)^2}{2!}[K_{AB}, [K_{AB}, K_P]] + \cdots, \quad (21)$$

where $\cdots$ contains nested commutators of higher orders. Thus, $K^{(+)}(s), K^{(-)}(s)$ only contains even and odd numbers of commutators respectively. Hence $K_P^{(+)}(s) \in S_+$ and $K_P^{(-)}(s) \in S_-$. Furthermore, using the cyclic property of the trace, we obtain $\text{Tr}(\rho_{AB} K_P(s)) = \text{Tr}(\rho_{AB} K_P)$, thus $\text{Tr}(\rho_{AB} K_P^{(-)}(s)) \equiv 0$ identically vanishes. $\square$

From the modular flow, we can construct a family of additive chirality measures, where two of the simplest examples are as follows.

**Corollary 3.4.** *The following two functions are additive chirality measures,*[7]

$$\gamma_s(\rho_{AB}) := i \, Tr(\rho_{AB}[K_A^{(+)}(s), K_B]), \quad (22)$$

$$\phi_s(\rho_{AB}) := i \, Tr(\rho_{AB}\{K_A^{(-)}(s), K_B\}), \quad (23)$$

*where* $s \in \mathbb{R}$.

Note that by taking derivatives with respect to $s$ at $s = 0$, one recovers previously constructed chirality measures:

$$\frac{d^2}{ds^2}\gamma_s(\rho_{AB})|_{s=0} = -J_3(\rho_{AB}) \quad (24)$$

$$\frac{d}{ds}\phi_s(\rho_{AB})|_{s=0} = -J_2(\rho_{AB}). \quad (25)$$

Note that each of the measures constructed so far in (16), (20), (22), (23), are odd under exchange of $A$ and $B$. It is also straightforward to construct measures without this symmetry property, such as $J_3' = i\text{Tr}(\rho_{AB}[[[K_{AB}, K_B], K_B], K_B])$.[8]

---

[7] Note that switching $A$ and $B$ in the definition of $\gamma_s$ and $\phi_s$ do not give new chirality measures. Taking the same subregion will not generate new measures either because $i\text{Tr}(\rho_{AB}[K_A^{(+)}(s), K_A])$ vanishes identically.

[8] One utility of such non-symmetric measures is that they can detect chirality in general "classical-quantum" states, which we will introduce in Sec. IV, in which $[K_{AB}, K_A] = 0$ but $[K_{AB}, K_B]$ is generally non-zero. A particular example of a chiral classical-quantum state is the one in Example 1, but in this fine-tuned case $J_3'$ is also zero.

The nested commutator construction also generalizes to multipartite quantum states. For a tripartite state, the simplest one is the modular commutator $J(A, B, C) = i\text{Tr}(\rho_{ABC}[K_{AB}, K_{BC}])$. One feature of the modular commutator is that it vanishes for tripartite pure states [7, 8], and therefore cannot detect chirality in these states. On the other hand, the bipartite chirality measures constructed in this section can detect chirality in tripartite pure states $|\psi_{ABC}\rangle$. One can trace out any of the three parties and compute the chirality measures, such as $\gamma_s$ or $\phi_s$, for the reduced density matrix of the remaining two parties. The tripartite pure state is chiral if one of these measures does not vanish.

## D. Chirality and entanglement in random two-qubit mixed states

Let us now use the chirality measures constructed above to detect chirality in the simple setup of random two-qubit mixed states. We randomly sample 50,000 states of the form $\rho_{AB} = U_{AB} D_{AB} U_{AB}^\dagger$, where $U_{AB}$ is a Haar-random unitary in $U(4)$, and $D_{AB}$ is a diagonal matrix $(p_1, p_2, p_3, p_4)$ uniformly chosen in $p_i \in (0, 1)$ with the constraint $\sum_i p_i = 1$ [9]. In order to contrast chirality with entanglement, we also compute the logarithmic negativity $E_N(\rho_{AB}) := \log \text{Tr}||\rho_{AB}^{T_B}||_1$ [32, 33]. It is well-known that the nonvanishing of $E_N$ is a necessary and sufficient condition for a two-qubit state to be entangled [34]. The result is shown in Fig. 2, where we find that $J_2$ is generically non-zero. Furthermore, there is no simple relation between entanglement and chirality, as many chiral states have negligible $E_N$ and many highly entangled states have negligible $|J_2|$. Indeed, the first example of a chiral state that we constructed in (2) is a separable state and hence has $E_N = 0$.

While chirality is not related to quantum entanglement, in the rest of the paper, we will show that chirality does give a lower bound on two other kinds of quantum resources, magic and discord-like quantum correlations.

---

[9] There is no unique measure to generate random mixed states. Thus, in general one has to be careful to draw any connections between physical quantities based on the plot, as the correlation may be an artificial effect of the ensemble. Here, however, our main observation is that chirality and entanglement are unrelated, for which the 50,000 samples in the plot suffice. We thank Josep Batle for pointing this out.

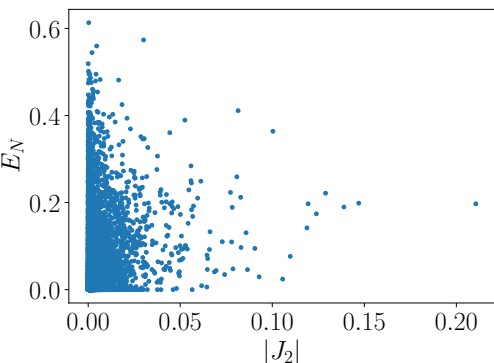

FIG. 2. Chirality measure $|J_2|$ versus logarithmic negativity $E_N$ in random two-qubit mixed states. The plot indicates that there is no simple relation between entanglement and chirality.

## III. CHIRALITY AND MAGIC

In this section, we relate the notion of chirality that we defined above to magic, a resource of quantum computation beyond the stabilizer formalism. We first prove that stabilizer states of qubits are nonchiral for any partition of the system. We then prove lower bounds on various measures of magic, including the stabilizer nullity and the stabilizer fidelity, in terms of the chiral log-distance.

### A. Stabilizer states of qubits are nonchiral

Stabilizer states are a subset of states of $n$ qubits which are efficiently classically simulable, even though they can in general be highly entangled [18]. Let us first review the definition of these states.

Recall that the Pauli group $G_n$ on $n$ qubits consists of all possible $n$-fold tensor products involving the one-qubit Pauli operators $\{I, X, Y, Z\}$, each accompanied by any of the four phases $\pm 1, \pm i$. Pure and mixed stabilizer states of $n$ qubits are defined as follows in terms of certain subsets of the Pauli group [18, 35, 36]:

1. A *pure* stabilizer state can be uniquely specified by the property that it is invariant under the action of some subgroup $S$ of $G_n$ generated by $n$ mutually commuting elements of $G_n$. We will refer to the generators of $S$ as $S_1, ..., S_n$. The generators $S_j$ must satisfy the following properties for a unique invariant pure state to exist: (i) they are independent, in the sense that we cannot obtain any of the $S_j$ as a product of some subset of the $S_\ell$'s for $\ell \neq j$ up to overall phases,

and (ii) $S_j^2 = I$ for each $j$ (which means that there is no overall factor of $i$ outside the string of Pauli matrices in $S_j$).

2. To define a *mixed* stabilizer state, we can consider a subgroup $S$ of $G_n$ with a set of $k$ generators $S_j$ for some $k < n$ which again satisfy properties (i) and (ii) from the previous point. Then there is a subspace of dimension $2^{n-k}$ that is invariant under the action of $S$, we will define the mixed stabilizer state corresponding to $S$ as the maximally mixed state on this subspace.

In both cases, given the set of generators $\{S_i\}_{i=1}^k$ for $k \leq n$, the associated stabilizer state can be written as

$$\rho[\{S_i\}] = \frac{1}{2^{n-k}} \prod_{i=1}^{k} \frac{I + S_i}{2} . \qquad (26)$$

The generators of $S$ can be specified (up to phases) by two matrices $M_Z, M_X \in \mathbb{F}_2^{k \times n}$ in the following way. If the $x$-th site of the Pauli string $S_j$ is $I, X, Z, Y$ (up to phases), then $(M_Z^{jx}, M_X^{jx}) = (0,0), (0,1), (1,0), (1,1)$, respectively. If we consider the combined $k \times 2n$ matrix $[M_Z \, M_X]$, its $j$-th row therefore gives a description of the generator $S_j$.

**Proposition 4.** *Any stabilizer state on $n$ qubits is $n$-partite nonchiral.*

*Proof.* Under complex conjugation in the computational basis, $Y \to -Y$ and the other Pauli matrices remain unchanged. Hence,

$$\rho[\{S_i\}]^* = \rho[\{(-1)^{n_i^{(Y)}} S_i\}] \qquad (27)$$

where $n_i^{(Y)}$ is the number of $Y$'s in the generator $S_i$, and is given by

$$n_i^{(Y)} = \sum_{x=1}^{n} M_Z^{ix} M_X^{ix} . \qquad (28)$$

The state $\rho[\{S_i\}]$ is $n$-partite nonchiral if there exists a local unitary $U = \otimes_{x=1}^n U_x$ such that

$$U S_i U^\dagger = (-1)^{n_i^{(Y)}} S_i \quad \forall i = 1, ..., k . \qquad (29)$$

Let us consider local unitaries where each $U_x$ is a Pauli matrix, described by binary vectors $v_Z \in \mathbb{F}_2^n$ and $v_X \in \mathbb{F}_2^n$ as follows: $U_x$ is $I, X, Z, Y$ if $(v_Z^x, v_X^x) = (0,0), (0,1), (1,0), (1,1)$ respectively. Under such unitary transformations, any Pauli string can only change by an overall sign. Explicitly, we have

$$U S_i U^\dagger = \prod_{x=1}^{n} (-1)^{(v_X^x M_Z^{ix} + v_Z^x M_X^{ix})} S_i \qquad (30)$$

The exponent $(v_X^x M_Z^{ix} + v_Z^x M_X^{ix})$ mod 2 determines whether the Pauli matrix $U_x$ commutes or anticommutes with the stabilizer $S_i$.

Combining (28), (29) and (30), we see that a stabilizer state is nonchiral if there exist two $n$-dimensional binary vectors $v_Z, v_X$ such that

$$\forall i, \quad \sum_{x=1}^n (v_X^x M_Z^{ix} + v_Z^x M_X^{ix}) = \sum_{x=1}^n M_Z^{ix} M_X^{ix} \pmod 2. \tag{31}$$

In matrix form, the above equation can be written as

$$\begin{bmatrix} M_Z & M_X \end{bmatrix} \begin{bmatrix} v_X \\ v_Z \end{bmatrix} = b, \tag{32}$$

where $b = \mathrm{diag}(M_Z M_X^T)$ is the diagonal part of the matrix $M_Z M_X^T$.

Recall that a solution to the linear equations $Mv = b$ always exists if the rows of $M$ are linearly independent. Recall that the $i$'th row of $[M_Z \ M_X]$, which we can label $r_i$, specifies the $i$-th generator $S_i$. One can check that if the rows are not linearly independent, i.e. if there exists some $i$ such that

$$r_i = \sum_{j \neq i} a_j r_j, \tag{33}$$

for some set of $a_j \in \mathbb{F}_2$, then the corresponding generators satisfy $S_i = \prod_{j \neq i} S_j^{a_j}$ (up to a possible phase). But this contradicts our requirement that the $k$ generators $S_i$ are independent. Hence, the rows of $M$ must be linearly independent.

The solution to Eq. (32) always exists for any stabilizer state $\rho$. This implies that there always exists a Pauli string $Q = \otimes_x Q_x \in G_n$ such that $Q\rho Q^\dagger = \rho^*$. Thus all stabilizer states are $n$-partite nonchiral. $\quad\square$

Recall that if a state is $n$-partite nonchiral, then it is also $(n-1)$-partite nonchiral if one combines two of the subsystems in the partition. We therefore immediately have the following corollary:

**Corollary 4.1.** *A Pauli stabilizer state is nonchiral with respect to any partition of the qubits.*

Let us now make an observation which will be important for the discussion of the next subsection. Let us specialize to the case of pure stabilizer states, $k = n$. In this case, the stabilizer group $S$ can be used to define a complete orthonormal basis of the Hilbert space, where each basis state is an eigenstate of each generator $S_j$, and we independently choose the eigenvalues of each $S_j$ to be $+1$ or $-1$. In other words, we can define an orthonormal basis $\{|\psi_\mathbf{s}\rangle\}$ with $\mathbf{s} \in \{1, -1\}^{\otimes n}$ such that $S_i|\psi_\mathbf{s}\rangle = s_i|\psi_\mathbf{s}\rangle$. Since the Pauli string $Q$ sends each $S_i$ to its complex conjugate, it also takes all the basis states $|\psi_\mathbf{s}\rangle$ to $|\psi_\mathbf{s}^*\rangle$ [10]. Thus, we have the following corollary.

**Corollary 4.2.** *Given an orthonormal basis $\{|\psi_\mathbf{s}\rangle\}$ specified by a stabilizer group with $n$ generators as defined above, any linear combination of the basis states*

$$|\psi\rangle = \sum_\mathbf{s} a_\mathbf{s} |\psi_\mathbf{s}\rangle \tag{34}$$

*with $a_\mathbf{s} \in \mathbb{R}$ are nonchiral. Furthermore, each $|\psi\rangle$ of the form (34) is related to its complex conjugation $|\psi^*\rangle$ by the same unitary transformation*

$$Q|\psi\rangle = |\psi^*\rangle. \tag{35}$$

*for some $Q \in G_n$, where $Q$ can be taken to be any solution to (32). Similarly, any convex combination*

$$\rho = \sum_\mathbf{s} p_\mathbf{s} |\psi_\mathbf{s}\rangle\langle\psi_\mathbf{s}| \tag{36}$$

*with probability distribution $\{p_\mathbf{s}\}$ is nonchiral.*

**Remark.** $Q$ is not unique. In fact, there are $2^n$ solutions to Eq. (32) because the rank of $[M_Z \ M_X]$ is $n$ and the number of unknown variables $v_Z^x$, $v_X^x$ valued in $\mathbb{F}_2$ is $2n$. One observation is that the rows of $[M_Z \ M_X]$ exactly give the nullspace of the $n \times 2n$ matrix $[M_Z \ M_X]$, due to fact that $M_Z M_X^T + M_X M_Z^T = 0$ (mod 2), which comes from the commutativity of the stabilizers. One can check that this implies that given any local unitary $Q_0$ corresponding to a particular solution to Eq. (32), the local unitary $Q = Q_0 \prod_{i=1}^n S_i^{a_i}$ for any $\mathbf{a} \in \mathbb{F}_2^n$ also corresponds to a solution. This gives the complete set of Pauli strings that can transform a pure stabilizer state to its complex conjugate in the computational basis.

**Remark.** One important caveat is that the proof only works for qubit systems and does not naively generalize to higher qudit dimensions. We leave the question of whether a qudit stabilizer state can be chiral to future work.

## B. Lower bound on magic from chirality

Going beyond stabilizer states is one of the crucial steps towards universal quantum computation. Thus,

---

[10] Strictly speaking, $QS_iQ^\dagger = S_i^*$ only implies that $Q|\psi_\mathbf{s}\rangle = e^{i\theta_\mathbf{s}}|\psi_\mathbf{s}^*\rangle$, where $e^{i\theta_\mathbf{s}}$ is a phase factor that may depend on s. We can redefine $|\tilde{\psi}_\mathbf{s}\rangle = e^{i\theta_\mathbf{s}/2}|\psi_\mathbf{s}\rangle$ such that $Q|\tilde{\psi}_\mathbf{s}\rangle = |\tilde{\psi}_\mathbf{s}^*\rangle$. We will therefore ignore the phase factor later on.

understanding the nature of nonstabilizerness provides insight into quantum advantage. In order to characterize the amount of nonstabilizerness, the resource theory of magic has been put forward, in which one specifies stabilizer states as free states and stabilizer protocols, including Clifford operations and measurements in the Pauli basis, as free operations. Quantities that satisfy the axioms of resource theories for this set of free states and free operations are called magic monotones [17, 20, 21, 23].

Three measures of magic that we will use below are the basis-minimized stabilizer asymmetry (BMSA) $A_\alpha$, the stabilizer nullity $\nu$ and the stabilizer fidelity $F$. $\nu$ is a magic monotone for both pure and mixed states [23], and $-\log F$ is equal to a magic monotone called the min-relative entropy of magic for pure states [21]. $A_\alpha$ for general $\alpha$ is a magic monotone for deterministic stabilizer protocols [24]. Let us review the definitions of these quantities for general pure states $|\psi\rangle$ of $n$ qubits:

*BMSA*: For any choice of a stabilizer group $S$ with $2^n$ elements, we can define an orthonormal basis $|\psi_\mathbf{s}\rangle$, as discussed above Corollary 4.2. Any pure state $|\psi\rangle$ can then be expanded in the orthonormal basis as follows:

$$|\psi\rangle = \sum_\mathbf{s} a_\mathbf{s} |\psi_\mathbf{s}\rangle, \quad a_\mathbf{s} \in \mathbb{C} \qquad (37)$$

The BMSA $A_\alpha$ defined in [24] is equivalent to the minimum of the $\alpha$-Renyi entropy of the probability distribution $\{|a_\mathbf{s}|^2\}$ over all choices of the stabilizer group $S$.

*Stabilizer nullity:* Let $\tilde{G}_n$ be the set of Pauli strings of $n$ qubits with the overall phases fixed to $+1$. Given any pure state $|\psi\rangle$, consider the subset $\tilde{S}$ of $\tilde{G}_n$ such that all $P \in \tilde{S}$ take definite values in the state, i.e., $\langle\psi|P|\psi\rangle = \pm 1$. The number of elements in $\tilde{S}$ is $2^k$ for some integer $k$, as $\tilde{S}$ is generated by some subset $S_1, ..., S_k \in \tilde{G}_n$ which must be mutually commuting in order to simultaneously have definite values in any $|\psi\rangle$. The stabilizer nullity is then defined as $\nu := n - k$. Note that $\nu = 0$ if and only if the state is a stabilizer state.

*Stabilizer fidelity*: The stabilizer fidelity of a pure state $|\psi\rangle$ is the maximal fidelity between $|\psi\rangle$ and any stabilizer state $|\phi\rangle$, $F := \max_\phi |\langle\phi|\psi\rangle|^2$.

Given that stabilizer states are non-chiral, it is natural to ask if there is a relation between the amount of chirality and the amount of magic in an arbitrary state. Indeed, as we will show, chirality gives a lower bound to magic in a precise sense. It is worth noting that chirality cannot give an upper bound to magic, since for instance a product of the magic state $|T\rangle = 1/\sqrt{2}(|0\rangle + e^{i\pi/4}|1\rangle)$ among different sites is always non-chiral.

For a $n$-qubit pure state $|\psi\rangle$, the natural generalization of the chiral log-distance in Eq. (7) to a measure of $n$-partite chirality is

$$C(|\psi\rangle) = -\log \max_{\{U_i\}} |\langle\psi^*| \otimes_i U_i |\psi\rangle|^2. \qquad (38)$$

where $U_i$ is a local unitary acting on site $i$. Note that $C$ is basis-independent. In order to compare with magic, it is useful to define a basis dependent quantity

$$C_P(|\psi\rangle) = -\log \max_{P \in G_n} |\langle\psi^*|P|\psi\rangle|^2, \qquad (39)$$

where the local unitary is restricted to a Pauli string in the Pauli group $G_n$ under a prescribed computational basis. Note that based on our discussion in the previous subsection, both quantities vanish for a stabilizer state. We are now ready to state our main result of this section:

**Proposition 5.** *The following inequality holds for an arbitrary pure state $|\psi\rangle$ of $n$ qubits:*

$$C \leq C_P \leq A_2 . \qquad (40)$$

*Using the fact that*

$$A_2 \leq \nu, \quad A_2 \leq -2\log F, \qquad (41)$$

*we can further use (40) to lower-bound the stabilizer nullity and the min-relative entropy of magic in terms of the chirality as follows:*

$$C \leq \nu, \quad C \leq -2\log F. \qquad (42)$$

*Each of the inequalities above is saturated for the case where $|\psi\rangle$ is a pure stabilizer state.*

*Proof.* It is clear that $C \leq C_P$ since a Pauli string is a product of on-site unitaries.

To relate $C_P$ to measures of non-stabilizerness, it will be useful to consider an arbitrary stabilizer group generated by $S_1, S_2, \cdots S_n$, and an associated orthonormal basis $\{|\psi_\mathbf{s}\rangle\}$, as defined above Corollary 4.2. Then we can find a Pauli string $Q_0$ such that for each $\mathbf{s}$, $Q_0|\psi_\mathbf{s}\rangle = |\psi_\mathbf{s}^*\rangle$. As noted in the remark under Corollary 4.2, $Q_0$ is not unique and can be multiplied with any of the $d := 2^n$ stabilizers.

The maximum of $|\langle\psi^*|P|\psi\rangle|^2$ over all $P$ is lower-bounded by the average of this overlap for all Pauli strings in the above set, i.e.,

$$\max_{P \in G_n} |\langle\psi^*|P|\psi\rangle|^2 \geq \frac{1}{d} \sum_{\mathbf{v} \in \mathbb{F}_2^n} |\langle\psi^*|Q_0 S_1^{v_1} S_2^{v_2} \cdots S_n^{v_n}|\psi\rangle|^2 \qquad (43)$$

Next, note that since $|\psi_\mathbf{s}\rangle$ forms an orthonormal basis, $|\psi\rangle$ can be expanded in this basis as in (37).

Now we can expand the right hand side of (43) in this basis,

$$\frac{1}{d} \sum_{\mathbf{v} \in \mathbb{F}_2^n} |\langle \psi^*|Q_0 S_1^{v_1} S_2^{v_2} \cdots S_n^{v_n}|\psi\rangle|^2 \qquad (44)$$

$$= \frac{1}{d} \sum_{\mathbf{v} \in \mathbb{F}_2^n} \left| \sum_{\mathbf{s},\mathbf{s}'} a_{\mathbf{s}} a_{\mathbf{s}'} \langle \psi_{\mathbf{s}'}^*|Q_0 S_1^{v_1} S_2^{v_2} \cdots S_n^{v_n}|\psi_{\mathbf{s}}\rangle \right|^2 \quad (45)$$

$$= \frac{1}{d} \sum_{\mathbf{v} \in \mathbb{F}_2^n} \left| \sum_{\mathbf{s},\mathbf{s}'} a_{\mathbf{s}} a_{\mathbf{s}'} s_1^{v_1} s_2^{v_2} \cdots s_n^{v_n} \langle \psi_{\mathbf{s}'}^*|Q_0|\psi_{\mathbf{s}}\rangle \right|^2 \quad (46)$$

$$= \frac{1}{d} \sum_{\mathbf{v} \in \mathbb{F}_2^n} \left| \sum_{\mathbf{s}} a_{\mathbf{s}}^2 s_1^{v_1} s_2^{v_2} \cdots s_n^{v_n} \right|^2 \qquad (47)$$

$$= \frac{1}{d} \sum_{\mathbf{s},\mathbf{s}'} a_{\mathbf{s}}^2 (a_{\mathbf{s}'}^*)^2 \prod_{i=1}^{n} (s_i s_i' + 1) \qquad (48)$$

$$= \sum_{\mathbf{s}} |a_{\mathbf{s}}|^4, \qquad (49)$$

where in (46), we apply the stabilizers to the basis states to get the eigenvalues; in (47), we apply $Q_0$ to the basis states to get the complex conjugated states, and in (48) we rearrange the sum in a way that gives $\delta_{\mathbf{s}\mathbf{s}'}$ in the summand.

Thus, we get an upper bound $C_P \leq H_2$ in terms of the second Renyi entropy $H_2$ of the expansion coefficients,

$$H_2(|a_{\mathbf{s}}|^2) := -\log \sum_{\mathbf{s}} |a_{\mathbf{s}}|^4. \qquad (50)$$

Note that the bound holds for any decomposition of the form Eq. (40), where the basis states $|\psi_{\mathbf{s}}\rangle$ are the simultaneous eigenstates of a complete set of stabilizers. This means that we can take the minimum of $H_2$ among all possible choices of basis (or equivalently all possible choices of generators $S_1, ..., S_n$ of the stabilizer group),

$$C_P(|\psi\rangle) \leq \min_{\{|\psi_{\mathbf{s}}\rangle\}} H_2(|\langle \psi_{\mathbf{s}}|\psi\rangle|^2) = A_2(|\psi\rangle), \qquad (51)$$

where the last equality is the definition of BMSA. This proves the inequality (1).

To prove (2), we need to show that $A_2$ is upper-bounded both by $\nu$ and by $-2\log F$. This is already proven in Ref. [24], but we will provide a proof for completeness. We will make use of the Renyi entropy inequalities,

$$H_2 \leq H_0, \quad H_2 \leq 2H_\infty, \qquad (52)$$

where $H_0$ and $H_\infty$ are the max entropy and min en-

tropy of a probability distribution, respectively. We have

$$H_0 = \log \left( \text{number of } \mathbf{s} \text{ such that } |a_{\mathbf{s}}|^2 \neq 0 \right), \quad (53)$$
$$H_\infty = -\log \max_{\mathbf{s}} |a_{\mathbf{s}}|^2 \qquad (54)$$

To show the inequality (1), suppose the $k$ independent stabilizers that take definite values are $S_1, S_2, \cdots S_k$. We can supplement this set with $\nu = n-k$ additional generators $S_{k+1}, S_{k+2}, \cdots S_n$ such that together, $S_1, ..., S_n$ form a complete set of stabilizers that specifies a basis $|\psi_{\mathbf{s}}\rangle$. In the expansion of $|\psi\rangle$ in this basis, the number of nonzero elements is $2^\nu$, as the $n - k$ eigenvalues $s_{k+1}, ..., s_n$ are not fixed in $|\psi\rangle$, while $s_1, ..., s_k$ are fixed. Thus from (53), $H_0 = \nu$ for this choice of $\{|\psi_{\mathbf{s}}\rangle\}$, and consequently

$$A_2 \leq H_2 \leq H_0 = \nu. \qquad (55)$$

To show the inequality (2), we consider the stabilizers $S_1', S_2', \cdots S_n'$ which specify the state $|\phi\rangle$ that maximizes the fidelity $F = |\langle \phi|\psi\rangle|^2$. This set of stabilizers also specify a complete basis $|\psi_{\mathbf{s}}'\rangle$. For the expansion of $|\psi\rangle$ in this basis, the min entropy is

$$H_\infty = -\log \max_{\mathbf{s}} |\langle \psi_{\mathbf{s}}'|\psi\rangle|^2 = -\log F. \qquad (56)$$

Thus, we have

$$A_2 \leq H_2 \leq 2H_\infty = -2\log F. \qquad (57)$$

Furthermore, for a stabilizer state, $C = C_P = A_2 = \nu = -2\log F = 0$, hence the inequalities are saturated. $\square$

Combining Propositions 2 and 5, one immediately gets the following corollary.

**Corollary 5.1.** *(Boundedness of chiral log-distance) Consider a quantum system on $n$ qubits, where $\{A_i\}_{i=1}^n$ represents a qubit each.*

*(1) For a pure state $|\psi_{A_1 A_2 \cdots A_n}\rangle$, the $n$-partite chiral log-distance satisfies that $C_{A_1 A_2 \cdots A_n}(|\psi\rangle) \leq n$.*

*(2) For a mixed state $\rho_{A_1 A_2 \cdots A_n}$, the $n$-partite chiral log-distance satisfies that $C_{A_1 A_2 \cdots A_n}(\rho) \leq 2n$*

*Proof.* For a pure state $|\psi_{A_1 A_2 \cdots A_n}\rangle$, we have $\nu \leq n$ which follows from the definition of stabilizer nullity. Thus, $C(|\psi\rangle) \leq \nu \leq n$ follows from Proposition 5.

For a mixed state $\rho_{A_1 A_2 \cdots A_n}$, one can purify it into a pure state of $2n$ qubits $|\tilde{\psi}_{A_1 A_1' A_2 A_2' \cdots A_n A_n'}\rangle$. Then the $2n$-partite chiral log-distance of $|\tilde{\psi}\rangle$ satisfies that $C(|\tilde{\psi}\rangle) \leq 2n$. By Proposition 2, we have $C_{A_1 A_2 \cdots A_n}(\rho) \leq C(|\tilde{\psi}\rangle) \leq 2n$. $\square$

We make several remarks. Firstly, Proposition 5 can be viewed as the "robust" version of Proposition 4. It

implies that if a state is slightly magical, then it cannot be too chiral. Secondly, while the computation of $C$ requires an optimization over continuous variables, the computation of $C_P$, $A_2$ or $F$ only involves optimization over a discrete (but exponentially large) set. Thirdly, Proposition 5 applies to generic pure states of qubits, including physical systems that describe chiral or nonchiral phases of matter. It would therefore be interesting to study the amount of chirality and the amount of magic in the physical systems to see how tight the bound is. Lastly, the bound can be used to derive bounds of other notions of magic from chirality, such as the number of $T$ gates needed to prepare the state in addition to Clifford operations. The applications to many-body physics will be explored in future work.

## IV.  CHIRALITY AND QUANTUM CORRELATIONS

In this section, we relate chirality to another resource known as "discord-like quantum correlations." We will review this notion of quantum correlations in Sec. IV A, discuss the relation of both chirality and discord-like correlations to non-commutativity of density matrices in Sec. IV B, and establish a lower-bound relating a particular chirality measure to a measure of discord-like correlations in Sec. IV C and IV D.

### A.  Review of discord-like quantum correlations

Recall that a separable or unentangled mixed state between two parties $A$ and $B$ takes the following form:

$$\rho_{AB}^{(\text{sep})} = \sum_i p_i\, \rho_A^{(i)} \otimes \rho_B^{(i)} \qquad (58)$$

for some probability distribution $\{p_i\}$ and some density matrices $\rho_A^{(i)}$, $\rho_B^{(i)}$. Such states are often seen as being "classically correlated," in the sense that they can be prepared using only local operations and classical communications between $A$ and $B$, with no need for EPR pairs between $A$ and $B$ in the initial state. However, Ref. [26] introduced a different notion of the distinction between classical and quantum correlations. According to their definition, $\rho_{AB}$ is classically correlated with respect to $A$ if there exists a local complete projective measurement in $A$ with some rank-1 measurement operators $\{\Pi_A^{(n)}\}_{n=1}^{d_A}$ such that the state $\rho_{AB}$ is unchanged by the measurement from the perspective of an observer without access to the measurement out-

come $n$, i.e.,

$$\sum_{n=1}^{d_A} \Pi_A^{(n)} \rho_{AB} \Pi_A^{(n)} = \rho_{AB}\,. \qquad (59)$$

A state satisfying the condition (59) can always be written in the form

$$\rho_{AB} = \sum_i p_i |i\rangle\langle i|_A \otimes \rho_B^{(i)}. \qquad (60)$$

for an orthonormal basis $\{|i\rangle_A\}$ and an arbitrary set of states $\rho_B^{(i)}$. Such states are referred to as *classical-quantum states*, or classically correlated states with respect to $A$. Any state which cannot be written in the form (60) is said to have discord-like quantum correlations with respect to $A$, even if it is an unentangled state as in (58). A further subset of (60)

$$\rho_{AB} = \sum_{i,j} p_{ij} |i\rangle\langle i|_A \otimes |j\rangle\langle j|_B, \qquad (61)$$

is said to be classically correlated (or just *classical*) with respect to both $A$ and $B$, where $\{|j\rangle_B\}$ is an orthonormal basis of $B$.

Discord-like quantum correlations have been quantified using a variety of measures, such as the quantum discord defined in [26], and the interferometric power (IP) defined in Refs. [28, 29]. Moreover, a resource-theoretic formulation has been developed for such correlations where the free states are the classical-quantum states in (60), and the free operations are arbitrary local channels in $B$ and local commutativity-preserving operations (LCPO) in $A$. See [27] for a review.

In this section, we will discuss the relation between chirality and discord-like quantum correlations. The key underlying idea is that the computable chirality measures that we introduced in Sec. II C are non-zero only if at least one of the commutators $[\rho_{AB}, \rho_A]$ or $[\rho_{AB}, \rho_B]$ is non-zero. The non-vanishing of such commutators is also related to discord-like quantum correlations, in ways that we will make precise in the next two subsections. In Section IV B, we will show that for generic states, the vanishing of $[\rho_{AB}, \rho_A]$ implies that $\rho_{AB}$ is a classical-quantum state, and use this as an intermediate step in showing that the vanishing of $[\rho_{AB}, \rho_A]$ implies that $\rho_{AB}$ is bipartite nonchiral under certain further conditions. In Sec. IV C, we will introduce a new measure of discord-like correlations called the intrinsic interferometric power, and in Sec. IV D, we will show that this quantity is lower-bounded by the chirality measure $\gamma_s$ introduced in (22).

However, it is important to note that the interplay between chirality and discord-like quantum correla-

tions is subtle, and is not fully captured by this inequality. In particular, while states that are classical with respect to both $A$ and $B$ as in (61) are clearly non-chiral, general classical-quantum states from (60) are not always non-chiral. Returning to our first example of a chiral quantum state in (2), we can see that it is an example of a classical state with respect to $A$. This fact is still consistent with the inequality we will derive, as the chirality of the state (2) is not detected by the measure $\gamma_s$.

## B. Chirality and noncommutativity

As we have shown in Sec. II, a large class of chirality measures are given by nested commutators. Thus, in order for a chiral state to be detected by these chirality measures, noncommutativity between the density matrix $\rho_{AB}$ and its marginals $\rho_A$ and $\rho_B$ is essential. This raises the following question: does chirality always require noncommutativity of the density matrices? Equivalently, is the state always nonchiral if the commutativity conditions $[\rho_{AB}, \rho_A] = 0$ and $[\rho_{AB}, \rho_B] = 0$ hold? Note that a stabilizer state satisfies the commutativity condition but is nonchiral. In the following, we will show that the answer to the question is yes given some additional constraints. We will need the following lemma.

**Lemma 2.** *For any bipartite state $\rho_{AB}$, if $\rho_A$ is nondegenerate and $[\rho_{AB}, \rho_A] = 0$, then $\rho_{AB}$ is a classical-quantum state.*

*Proof.* Recall that the existence of a rank-1 projective measurement in $A$ that leaves the state invariant (eq. (59)) is equivalent to $\rho_{AB}$ being a classical-quantum state (60). If $[\rho_{AB}, \rho_A] = 0$ and $\rho_A$ is non-degenerate, the eigenstates of $\rho_A$ provide such a measurement basis.

Alternatively, we can directly verify the above statement as follows. Let

$$\rho_A = \sum_i p_i |i\rangle\langle i| \tag{62}$$

be the eigenvalue decomposition of $\rho_A$, where $p_i$'s are non-degenerate. We can expand $\rho_{AB}$ in this basis

$$\rho_{AB} = \sum_{i,j} |i\rangle\langle j| \otimes \sigma_{ij} \tag{63}$$

Now we can compute

$$[\rho_{AB}, \rho_A] = \sum_{i,j} (p_i - p_j)|i\rangle\langle j| \otimes \sigma_{ij} \tag{64}$$

If the commutator vanishes, it implies each term in the

above sum vanishes. Due to the nondegeneracy of $p_i$'s, the cross term vanishes, thus,

$$\sigma_{ij} = p_i \rho_i \delta_{ij}. \tag{65}$$

Thus,

$$\rho_{AB} = \sum_i p_i |i\rangle\langle i| \otimes \rho_i \tag{66}$$

is a classical-quantum state. $\square$

Next, we prove that chirality requires noncommutativity of either $\rho_{AB}$ and $\rho_A$ or $\rho_{AB}$ and $\rho_B$, given some additional conditions.

**Proposition 6.** *For a bipartite state $\rho_{AB}$, if $[\rho_{AB}, \rho_A] = 0$ and $[\rho_{AB}, \rho_B] = 0$, then $\rho_{AB}$ is a nonchiral state if given one of the following three conditions:*
*(1) Both $\rho_A$ and $\rho_B$ are nondegenerate.*
*(2) $\rho_A$ is nondegenerate and $A$ is a qubit.*
*(3) $\rho_{AB}$ is a two-qubit state.*

*Proof. About condition (1):* Now suppose $\rho_A$ and $\rho_B$ are both nondegenerate. Then by Lemma 2, the state $\rho_{AB}$ is a classical-quantum state and $\rho_B = \sum_i p_i \rho_i$. We can use $[\rho_{AB}, \rho_B] = 0$, which implies that

$$[\rho_B, \rho_i] = 0, \tag{67}$$

which indicates that $\rho_B$ and $\rho_i$ are simultaneously diagonalizable. If $\rho_B$ is nondegenerate, then the eigenvalue decomposition is *unique*. Thus $\rho_i$ can be diagonalized in the *same* basis,

$$\rho_i = \sum_j q_{ij}|j\rangle\langle j|. \tag{68}$$

This implies that the state $\rho_{AB}$ is a classical ensemble,

$$\rho_{AB} = \sum_{i,j} p_i q_{ij}|i\rangle\langle i| \otimes |j\rangle\langle j| \tag{69}$$

which is nonchiral. This completes the first part of the proof.

*About condition (2):* Now suppose $A$ is a qubit, and $\rho_A$ is nondegenerate. We can use Lemma 2 to restrict $\rho_{AB}$ to a classical-quantum state. Since $A$ is a qubit, we have $\rho_B = p_0 \rho_0 + p_1 \rho_1$, where $0 < p_0, p_1 < 1$. Then Eq. (67) implies that

$$[\rho_0, \rho_1] = 0. \tag{70}$$

Thus again they are simultaneously diagonalizable and the state is of the form Eq. (69). The same argument applies to the case that $\rho_B$ is nondegenerate.

*About condition (3):* Now we consider the case $A, B$ are two qubits. The only remaining case is that both

$\rho_A$ and $\rho_B$ are degenerate, which in the qubit case reduces to the conditions that $\rho_A = I/2, \rho_B = I/2$. We can then expand

$$\rho_{AB} = \frac{1}{4}\mathbb{I} + \sum_{i,j=1}^{3} \beta_{ij}\sigma^i \otimes \sigma^j, \qquad (71)$$

where $\sigma^i$'s are Pauli matrices. Now we use the result of Ref. [37], which asserts that two two-qubit mixed states can be connected by local unitaries if and only if all the 18 invariants defined in Ref. [37] are equal. If the two marginals are maximally mixed, there are only 3 nonvanishing invariants among the 18, which are

$$I_1 = \det(\beta), I_2 = \mathrm{Tr}(\beta^T \beta), I_3 = \mathrm{Tr}((\beta^T \beta)^2). \quad (72)$$

Under complex conjugation, $\beta$ transforms by an orthogonal matrix,

$$\beta \to O\beta O^T, \quad O = \mathrm{diag}(1, -1, 1). \qquad (73)$$

Thus all the three invariants do not change. The result of Ref. [37] then implies that $\rho$ and $\rho^*$ are related by local unitaries. Thus $\rho_{AB}$ is nonchiral. $\qquad\square$

Going beyond two qubits, however, it is possible to construct a chiral state which satisfies the commutativity condition.

**Example 2.** *The following qudit-qubit state $\rho_{AB}$ satisfies $[\rho_{AB}, \rho_A] = 0$ and $[\rho_{AB}, \rho_B] = 0$ and is chiral. Let $\mathcal{H}_A$ be 4-dimensional and B be a qubit.*

$$\rho_{AB} = \sum_{i=1}^{3} p_i |i\rangle_A \langle i| \otimes |\psi_i\rangle_B \langle\psi_i| + p_4 |4\rangle\langle 4| \otimes \rho_4 \quad (74)$$

*where $|\psi_i\rangle$'s are identical to those in Eq (3) and the state $\rho_4$ satisfies*

$$\sum_{i=1}^{3} p_i |\psi_i\rangle\langle\psi_i| + p_4\rho_4 = I/2 \qquad (75)$$

*and $p_i$'s are nondegenerate and nonzero. Note that the solution of $\rho_4$ to Eq. (75) can always be found if $p_1, p_2, p_3$ are sufficiently small.*

$\rho_{AB}$ is chiral by the same argument given for the example in Eq. (2). It is worth noting that this counterexample requires the degeneracy of $\rho_B$. This is a consequence of Proposition 6, which indicates that such counterexamples have to be fine-tuned. The chirality of this state cannot be detected by any of the nested commutator ansatz. Nevertheless, it is still a chiral state by definition and can be detected by the chiral log-distance $C_{A|B}(\rho) > 0$.

To summarize, given a bipartite quantum state $\rho_{AB}$ with nondegenerate $\rho_A$ and $\rho_B$, a chiral state requires noncommutativity, that is $[\rho_{AB}, \rho_A] \neq 0$ or $[\rho_{AB}, \rho_B] \neq 0$. Thus, state $\rho_{AB}$ changes under a local measurement in the basis of $\rho_A$ or $\rho_B$. This implies that a chiral state generically has discord-like quantum correlations. We will make this statement more quantitative in the next subsection.

## C. Intrinsic interferometric power as a measure of quantum correlations

In this section, we will motivate and define a new measure of discord-like quantum correlations involving a certain quantum Fisher information, closely related to the interferometric power. Unlike the interferometric power, we will define this quantity in terms of intrinsic properties of the state rather than with reference to some externally given Hamiltonian. We will show that this measure of quantum correlations is lower-bounded by the chirality measure in (22), recalled here for clarity

$$\gamma_s(\rho_{AB}) = i\mathrm{Tr}(\rho_{AB}[K_A^{(+)}(s), K_B]), \qquad (76)$$

where $K_A^{(+)}(s) = (\rho_{AB}^{is}K_A\rho_{AB}^{-is} + \rho_{AB}^{-is}K_A\rho_{AB}^{is})/2$. While this does not guarantee that all chiral states have quantum correlations (notably, examples of chiral classical-quantum states such as Example 1 have zero values both of all our chirality measures and of the quantum Fisher information), it does indicate that states whose chirality can be detected by the measure in (22) have non-trivial quantum correlations.

Given a state $\rho$ and a Hamiltonian $H$, it can be associated with the one-parameter family of states $\rho(H, s) = e^{iHs}\rho e^{-iHs}$, where $s \in \mathbb{R}$. The sensitivity of this family of states to the parameter $s$ is quantified by the the quantum Fisher information (QFI), which is defined as follows:

$$F_H(\rho(s)) = \mathrm{Tr}\left(\frac{\partial\rho}{\partial s}\mathcal{R}_{\rho(s)}^{-1}\left(\frac{\partial\rho}{\partial s}\right)\right)$$

$$= -\mathrm{Tr}\left([H, \rho(s)]\mathcal{R}_{\rho(s)}^{-1}([H, \rho(s)])\right) \quad (77)$$

Here $\mathcal{R}_\rho^{-1}$ is a superopererator whose action on any operator $O$ is defined as follows in terms of the eigenstates $\{|\psi_i\rangle\}$ and eigenvalues $p_i$ of $\rho$: [38]

$$\mathcal{R}_\rho^{-1}(O) = \sum_{\substack{j,k \text{ such that} \\ p_j+p_k \neq 0}} \frac{2}{p_j + p_k} \langle\psi_j|O|\psi_k\rangle |\psi_j\rangle\langle\psi_k|$$

$$(78)$$

The above operator is a solution to the equation

$$R_\rho^{-1}(O)\rho + \rho R_\rho^{-1}(O) = 2O \qquad (79)$$

and its action on $\frac{\partial \rho}{\partial s}$ is known as the symmetric logarithmic derivative. For our later discussion, it will be useful to note that if $\rho$ is full-rank, then

$$R_\rho^{-1}(O) = \int_{-\infty}^{\infty} ds \frac{1}{\cosh \pi s} \rho^{-\frac{1}{2}+is} O \rho^{-\frac{1}{2}-is}. \qquad (80)$$

This identity can be readily verified by expressing the RHS of (80) in the eigenbasis of $\rho$ and using the fact that $\int_{-\infty}^{\infty} ds \frac{1}{\cosh \pi s} e^{iks} = \frac{1}{\cosh(k/2)}$.

The QFI has found wide applications to quantum metrology and estimation theory due to the Cramer-Rao bound, according to which the QFI determines the minimum uncertainty in estimates of the parameter $s$ from measurements of the state $\rho$ [38–40].

In what follows, we focus on a bipartite system $\rho_{AB}$, and take the Hamiltonian to be the modular Hamiltonian of one subsystems, $H = K_i := \log \rho_i$. The corresponding quantum Fisher information is denoted as $F^{(i)}(\rho_{AB})$,

$$F^{(i)}(\rho_{AB}) := F_{K_i}(\rho_{AB}), \qquad (81)$$

which in the close form reads $F^{(i)}(\rho_{AB}) = -\text{Tr}(\rho_{AB}(R_{\rho_{AB}}^{-1}([K_i, \rho_{AB}]))^2)$, where $i$ could be $A$ or $B$.

We will call this quantity intrinsic interferometric power due to its close relation with the interferometric power (IP) defined in Refs. [28, 29]. Given a bipartite state $\rho_{AB}$ and a real diagonal matrix $\Gamma_A$ of dimension $d_A$, the IP is defined by the minimal quantum Fisher information with respect to a Hamiltonian on $A$ with the same spectrum as $\Gamma_A$, i.e.,

$$\mathcal{P}_{\Gamma_A}(\rho_{AB}) = \min_{H_A \sim \Gamma_A} F_{H_A}(\rho_{AB}), \qquad (82)$$

where $\sim$ means that the eigenvalues are the same. The IP is a resource of quantum correlations because it satisfies the set of axioms in Ref. [27]. These includes the four properties listed in Table I. The intrinsic IP defined in Eq. (81) satisfies a similar set of properties, with important differences explained below. The proof is straightforward, see appendix for more details.

Let us clarify a few of the properties in the table. (1) Faithfulness means that the quantity vanishes and only vanishes for classical-quantum states. The IP is faithful if $\Gamma_A$ is nondegenerate. This is also true for the intrinsic IP if $\rho_A$ is nondegenerate. (2) Local unitary invariance means that the quantity is invariant under the conjugation of $U_A$ and $U_B$. This is satisfied for both quantities. (3) Monotonicity means that the

quantity is monotonically decreasing under a quantum channel on $B$. This is also satisfied for both quantities. Note the apparent asymmetry between the subsystems $A$ and $B$ in the properties (1) and (3). One may also define another IP $P_{\Gamma_B}(\rho_{AB})$ and another intrinsic IP $F^{(B)}(\rho_{AB})$ that satisfy similar properties with the roles of $A$ and $B$ reversed. The property (4) is where the IP and the intrinsic IP differ most. For pure states, the IP reduces to minimal local variance [28], an entanglement monotone. However, the intrinsic IP reduces to capacity of entanglement [41], that is $\langle K_A^2 \rangle - \langle K_A \rangle^2$. It is not an entanglement monotone as it vanishes for maximally entangled states. This is closely related to the property (1) of the intrinsic IP, which requires nondegeneracy of $\rho_A$ for faithfulness.

### D. Lower bound on intrinsic interferometric power from chirality

We will now show that the intrinsic IP can be lower bounded by one of our nested commutator chirality measures from Sec. II C. Specifically, we integrate Eq. (76) with respect to the modular flow parameter $s$ to obtain an additive chirality measure

$$\gamma(\rho_{AB}) = \int_{-\infty}^{\infty} ds \frac{1}{\cosh \pi s} \gamma_s(\rho_{AB}). \qquad (83)$$

**Proposition 7.** *The following two inequalities hold if $\rho_{AB}$ is full-rank:*

$$|\gamma(\rho_{AB})|^2 \le Tr(\rho_A K_A^2) F^{(B)}(\rho_{AB}). \qquad (84)$$

$$|\gamma(\rho_{AB})|^2 \le Tr(\rho_B K_B^2) F^{(A)}(\rho_{AB}). \qquad (85)$$

*Proof.* Using the cyclic property of trace, we can rewrite Eq. (76) as

$$\gamma_s = -i\text{Tr}(K_A^{(+)}(s)[\rho_{AB}, K_B]) \qquad (86)$$

Combining Eqs. (86) and (83), together with the fact that $1/\cosh(\pi s)$ is an even function of $s$, we see that

$$\gamma(\rho_{AB}) = -\int_{-\infty}^{\infty} ds \frac{1}{\cosh \pi s} \text{Tr}(K_A \rho_{AB}^{is}[\rho_{AB}, K_B] \rho_{AB}^{-is}) \qquad (87)$$

Using (80), we then have

$$\gamma(\rho_{AB}) = -i\text{Tr}(K_A \rho_{AB}^{1/2} R_{\rho_{AB}}^{-1}([\rho_{AB}, K_B]) \rho_{AB}^{1/2}) \qquad (88)$$

We can now use the Cauchy Schwarz inequality

$$|\text{Tr}(PQ)|^2 \le \text{Tr}(P^\dagger P)\text{Tr}(Q^\dagger Q), \qquad (89)$$

with $P = K_A \rho_{AB}^{1/2}$, $Q = R_{\rho_{AB}}^{-1}([\rho_{AB}, K_B])\rho_{AB}^{1/2}$. Next,

| Property | IP $P_{\Gamma_A}(\rho_{AB})$ | Intrinsic IP $F^{(A)}(\rho_{AB})$ |
|---|---|---|
| 1) Faithfulness | is faithful if $\Gamma_A$ is nondegenrate | is faithful if $\rho_A$ is nondegenerate |
| 2) Local unitary invariance | is invariant under local unitaries | is invariant under local unitaries |
| 3) Monotonicity | is monotonically decreasing under local quantum channels on $B$ | is monotonically decreasing under local quantum channels on $B$ |
| 4) Reduction for pure states | reduces to minimal local variance [28] which is an entanglement monotone | reduces to capacity of entanglement [41] which is NOT an entanglement monotone |

TABLE I. Properties of the IP and the intrinsic IP.

note that

$$\mathrm{Tr}(P^\dagger P) = \mathrm{Tr}(\rho_{AB}K_A^2) = \mathrm{Tr}(\rho_A K_A^2) \qquad (90)$$

and

$$\begin{aligned}
\mathrm{Tr}(Q^\dagger Q) &= -\mathrm{Tr}(\rho_{AB}(R_{\rho_{AB}}^{-1}([\rho_{AB}, K_B]))^2) \\
&= -\mathrm{Tr}\left([\rho_{AB}, K_B] R_{\rho_{AB}}^{-1}([\rho_{AB}, K_B])\right) \\
&= F^{(B)}(\rho_{AB}). \qquad (91)
\end{aligned}$$

Then Eq. (89) implies that

$$|\gamma(\rho_{AB})|^2 \leq \mathrm{Tr}(\rho_A K_A^2) F^{(B)}(\rho_{AB}) \qquad (92)$$

as desired.

Recall from Sec. II C that $\gamma_s$ is odd under exchange of $A$ and $B$. The second inequality in the statement of the proposition thus follows from the same proof with the roles of $A$ and $B$ exchanged. $\qquad \square$

We note that the prefactor $\mathrm{Tr}(\rho_i K_i^2)$ can be bounded above with the dimension $d_i$ of the subsystem $i = A, B$.

**Corollary 7.1.**

$$|\gamma(\rho_{AB})|^2 \leq c(d_i) F^{(j)}(\rho_{AB}), \qquad (93)$$

where $(i,j) = (A, B)$ or $(B, A)$ and $c(d) = (\log d)^2$ for $d \geq 3$.

The proof is simple and can be found in Appendix B.

Let us comment on the implication of the proposition above. Proposition 7 indicates that if the chirality measure $\gamma(\rho_{AB}) \neq 0$, then both quantum Fisher information $F^{(A)}(\rho_{AB}) > 0$ and $F^{(B)}(\rho_{AB}) > 0$. Thus, a chiral state whose chirality can be detected by the measure in (83) must have non-trivial quantum correlations. Note that, however, $\gamma(\rho_{AB}) = 0$ does not imply that the state is nonchiral. As noted earlier, the chiral classical-quantum in Eq. (2) is trivially consistent with Proposition 7 despite being chiral, as it has both $F^{(A)} = 0$ and $\gamma(\rho_{AB}) = 0$.

## V. DISCUSSION

In this work, we have discussed the notion of chirality for generic quantum states. We defined a chiral state as one that cannot be connected with its complex conjugate in a local product basis by local unitary transformations. We introduced two types of chirality measures. We first considered a measure called the chiral log-distance, which is operationally well-motivated and even under complex conjugation. We further constructed a series of additive chirality measures based on a nested commutator ansatz, which are odd under complex conjugation and additive under tensor product. The non-vanishing of any such measure can be used as an indicator of chirality in a given state.

We showed the relation between chirality and two kinds of quantum resources, magic and discord-like quantum correlations. We showed that all stabilizer states of qubits are non-chiral, and further that the chiral log-distance lower-bounds various measures of magic, including the BMSA with $\alpha = 2$, the stabilizer nullity and the min-relative entropy of magic. We then discussed the relation between chirality and discord-like quantum correlations. We showed that chirality requires noncommutativity between the full system and subsystem reduced density matrices under certain generic conditions. More quantitatively, we introduced the intrinsic interferometric power as a measure of quantum correlations, and showed a lower bound on this quantity in terms of one of the additive nested commutator chirality measures.

One important question to resolve in future work is whether one can construct a closed-form expression in term of the density matrix which serves as a faithful chirality measure. There are bipartite chiral states whose chirality cannot be detected by the additive measures we construct by the nested commutators (Example 2). Even though the chiral log-distance can always detect such chirality, it is not clear whether there is a computation-efficient measure without optimization over unitaries. Understanding the physics of this exotic type of chirality remains a future problem.

Our work opens up several future directions. We found that chirality gives a lower bound on magic,

which hinders classical simulation of quantum dynamics. It is then a natural question whether one can more directly understand why chirality provides an obstacle to classical simulation. If a state is chiral, then there is no local basis such that the wavefunction is real, which indicates the sign problem in a Monte-Carlo simulation. A related question is whether chiral states can be efficiently represented by tensor networks, such as matrix product states or projected entangled pair states.

Another interesting direction would be to compute the various chirality measures introduced in this paper in quantum many-body states, and use them to identify universal properties of chiral quantum phases. It would further be interesting to understand how these measures evolve under quantum dynamics, such as Clifford circuits doped with T gates, floquet dynamics, chaotic evolutions, and ergodicity-breaking systems. The behaviour of magic has previously been studied in these contexts, and it would be interesting to compare it to the behaviour of chirality. One particularly interesting example is the generalized $W$ state in one dimension [42], which exhibits both an extensive amount of chirality and magic.

As discussed in the introduction and Sec. II A, $n-$partite chirality with a fixed partition falls out of the scope of conventional resource theories due to its non-monotonicity under partial trace. The fact that additive chirality measures are necessarily odd under time reversal, as we discussed in Sec. II B, and hence can have either sign, poses a further challenge to formulating a resource theory. However, this unusual feature of oddness can be viewed as a benefit rather than a drawback, as it seems to detect a certain geometry of the multipartite quantum state that is invisible to the traditional resource measures. The measure of quantum state geometry modulo local unitary resonates with those considered in few-qubit states in the early days [37, 43, 44]. It would be interesting to make this geometric picture more precise.

From a more fundamental perspective, chirality underlies the importance of complex numbers of quantum mechanics. It would be thus interesting to relate chirality to the discrimination of real and complex quantum mechanics, which several recent theoretical proposals and experiments have demonstrated [45–47].

## ACKNOWLEDGMENTS

We thank Hilary Carteret for helpful discussions and, in particular, for pointing us to Ref. [37]. We thank Tyler Ellison, Salvatore Francesco Emanuele Oliviero, Jovan Odavic, Joseph Batle, Poetri Sonya Tarabunga and Beni Yoshida for useful discussions on the manuscript. This work was supported by the Perimeter Institute for Theoretical Physics (PI) and the Natural Sciences and Engineering Research Council of Canada (NSERC). Research at PI is supported in part by the Government of Canada through the Department of Innovation, Science and Economic Development Canada and by the Province of Ontario through the Ministry of Colleges and Universities. SV is supported by Google and SITP. IK and BS acknowledge support from NSF under award number PHY-2337931. BS was supported by the Simons Collaboration on Ultra-Quantum Matter, a grant from the Simons Foundation (652264, JM), the faculty startup grant of J. Y. Lee, and the IQUIST fellowship at UIUC.

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

## Appendix A: Proof of Proposition 3

In this appendix, we provide the technical details of proving Proposition 3.

*Proof.* For (1), the additivity follows from the fact that $\log \rho_{AA'BB'} = \log \rho_{AB} + \log \rho_{A'B'}$ for product state $\rho_{AA'BB'} = \rho_{AB} \otimes \rho_{A'B'}$. The action of complex conjugation is $(\log \rho_{AB})^* = \log \rho_{AB}^* = \log \rho_{AB}^T = (\log \rho_{AB})^T$. $\log \rho_A$ and $\log \rho_B$ follow from the same argument, (2) is obvious.

For (3), the additivity follows from

$$
\begin{aligned}
& [X(\rho_{AB} \otimes \rho_{A'B'}), Y(\rho_{AB} \otimes \rho_{A'B'})] \\
&= [X(\rho_{AB}) + X(\rho_{A'B'}), Y(\rho_{AB}) + Y(\rho_{A'B'})] \\
&= [X(\rho_{AB}), Y(\rho_{AB})] + [X(\rho_{A'B'}), Y(\rho_{A'B'})],
\end{aligned}
\tag{A1}
$$

where we have used the fact that operators on $AB$ and operators on $A'B'$ commute. The action of complex conjugation follows from

$$
\begin{aligned}
& [X(\rho_{AB}^*), Y(\rho_{AB}^*)] \\
&= [aX(\rho_{AB})^T, bY(\rho_{AB})^T] \\
&= ab[X(\rho_{AB})^T, Y(\rho_{AB})^T] \\
&= -ab[X(\rho_{AB}), Y(\rho_{AB})]^T.
\end{aligned}
\tag{A2}
$$

For (4), the additivity of the chirality measure follows from

$$
\begin{aligned}
& J(\rho_{AB} \otimes \rho_{A'B'}) \\
&= i\text{Tr}(\rho_{AB} \otimes \rho_{A'B'}(X(\rho_{AB}) + X(\rho_{A'B'}))) \\
&= i(\text{Tr}(\rho_{AB}X(\rho_{AB}))\text{Tr}(\rho_{A'B'}) + i\text{Tr}(\rho_{A'B'}X(\rho_{A'B'}))\text{Tr}(\rho_{AB}) \\
&= J(\rho_{AB}) + J(\rho_{A'B'}).
\end{aligned}
\tag{A3}
$$

Furthermore, $J(\rho_{AB})^* = -J(\rho_{AB}^*) = -i\text{Tr}(\rho_{AB}^* X(\rho_{AB}^*)) = i\text{Tr}(\rho_{AB}^T X(\rho_{AB})^T) = i\text{Tr}(\rho_{AB}X(\rho_{AB})) = -J(\rho_{AB})$. Thus, $J$ is real and odd under complex conjugation.

For (5), the additivity follows from

$$
\begin{aligned}
& J(\rho_{AB} \otimes \rho_{A'B'}) \\
&= i\text{Tr}\left(\rho_{AB} \otimes \rho_{A'B'} \{X(\rho_{AB}) + X(\rho_{A'B'}), Y(\rho_{AB}) + Y(\rho_{A'B'})\}\right) \\
&= i\text{Tr}\left(\rho_{AB} \{X(\rho_{AB}), Y(\rho_{AB})\}\right) + i\text{Tr}\left(\rho_{A'B'} \{X(\rho_{A'B'}), Y(\rho_{A'B'})\}\right) \\
&\quad + 2i\text{Tr}\left(\rho_{AB}X(\rho_{AB})\right)\text{Tr}\left(\rho_{A'B'}Y(\rho_{A'B'})\right) \\
&\quad + 2i\text{Tr}\left(\rho_{AB}Y(\rho_{AB})\right)\text{Tr}\left(\rho_{A'B'}X(\rho_{A'B'})\right) \\
&= i\text{Tr}\left(\rho_{AB} \{X(\rho_{AB}), Y(\rho_{AB})\}\right) + i\text{Tr}\left(\rho_{A'B'} \{X(\rho_{A'B'}), Y(\rho_{A'B'})\}\right) \\
&= J(\rho_{AB}) + J(\rho_{A'B'}).
\end{aligned}
\tag{A4}
$$

The action of complex conjugation follows from

$$
\begin{aligned}
J(\rho_{AB})^* &= -J(\rho_{AB}^*) = -i\mathrm{Tr}\left(\rho_{AB}^* \{X(\rho_{AB}^*), Y(\rho_{AB}^*)\}\right) \\
&= -\mathrm{Tr}\left(\rho_{AB}^T \{aX(\rho_{AB})^T, bY(\rho_{AB})^T\}\right) \\
&= -iab\mathrm{Tr}\left(\rho_{AB}^T \{X(\rho_{AB}), Y(\rho_{AB})\}^T\right) \\
&= -iab\mathrm{Tr}\left(\rho_{AB} \{X(\rho_{AB}), Y(\rho_{AB})\}\right) \\
&= J(\rho_{AB}).
\end{aligned}
\tag{A5}
$$

Thus, $J$ is an additive chirality measure. $\qquad\qquad\qquad\qquad\qquad\qquad\qquad\qquad\qquad\qquad\qquad\quad$ $\square$

## Appendix B: Bounding $\mathrm{Tr}(\rho_i K_i^2)$

In Corollary 7.1, we have used the upper bound $\mathrm{Tr}(\rho_i K_i^2) \leq c(d_i)$ where $c(d_i) = (\log d_i)^2$ for $d_i \geq 3$. In this appendix, we provide a simple proof. Let $0 < x_i < 1$ and $\sum_i x_i = 1$ be the eigenvalues of $\rho$, we can compute the maximum of

$$
f(x) = \sum_i x_i (\log x_i)^2
\tag{B1}
$$

using the Lagrange multiplier. Define

$$
F(x, \lambda) = f(x) + \lambda\left(\sum_i x_i - 1\right)
\tag{B2}
$$

The extrema are given by the solutions of

$$
\frac{\partial F}{\partial x_i} = 0, \quad \frac{\partial F}{\partial \lambda} = 0.
\tag{B3}
$$

This gives

$$
(\log x_i)^2 + 2\log x_i = -\lambda
\tag{B4}
$$

If $\lambda \leq 0$, then the only solution is all $x_i$ being equal, thus the maximum is given by $x_i = 1/d$ and $f(x) = (\log d)^2$. If $\lambda > 0$, then the two solutions of $\log x_i$ both have $-2 < \log x_i < 0$. Taking into account that $\sum_i x_i = 1$, we must have $d < e^2$, which means $d \leq 7$. Thus for $d \geq 8$ the only extremum is given by $x_i = 1/d$ for all $i$'s.

$$
f(x) \leq (\log d)^2, \quad d \geq 8.
\tag{B5}
$$

For $d \leq 7$, there are possibly many extrema, but one can enumerate them and compare the value of $f(x)$ case by case.

$d = 2$ - the extrema is either $x_1 = x_2 = 1/2$, in which case $f(x) = (\log 2)^2 = 0.4804$ or $x_1, x_2$ satisfying $x_1 x_2 = e^{-2}$ and $x_1 + x_2 = 1$. In the latter case one solution is given by $x_1 = 0.839, x_2 = 0.161$, which gives $f(x) = 0.563$. Thus the maximum is given by $x_1 = 0.839, x_2 = 0.161$.

$d = 3$ - The extrema is either $x_i = 1/3$, in which case $f(x) = (\log 3)^2 = 1.207$, or $x_1 = x_2 \neq x_3$. In the latter case we must have $x_1 x_3 = e^{-2}$. However, this contradicts the condition that $2x_1 + x_3 = 1$, as $2x_1 + x_3 \geq 2\sqrt{2x_1 x_3} = 2\sqrt{2}/e > 1$. Thus the only extremum is given by all $x_i = 1/3$ and $f(x) = (\log 3)^2 = 1.207$.

$d \geq 4$ - the only possible extrema is again all $x_i = 1/d$, analogous to the case of $d = 3$. For contradiction, consider if one $x_i := a$ is different from others and suppose there are $m$ of them which takes the same value, then the rest of $x_i$'s must take on the values $b = e^{-2}/a$. This indicates that $ma + (d - m)b = 1$, which is not possible since $ma + (d - m)b \geq 2\sqrt{m(d - m)ab} = 2\sqrt{m(d - m)}/e \geq 2\sqrt{d - 1}/e > 1$.

Thus we obtain $c(d) = \max f(x)$ for all $d$,

$$
c(d) = (\log d)^2, \quad d \geq 3.
\tag{B6}
$$

and $c(2) = 0.563 > (\log 2)^2$.

## Appendix C: Properties of the intrinsic IP

In the main text, we list four properties of the intrinsic IP, that is, the quantum Fisher information $F^{(A)}(\rho_{AB})$.

(1) If $\rho_{AB}$ is a classical-quantum state then $F^{(A)}(\rho_{AB})$ vanishes; furthermore if $\rho_A$ is nondegenerate then $F^{(A)}(\rho_{AB})$ vanishes only on classical-quantum states;

(2) $F^{(A)}(\rho_{AB})$ is invariant under local unitary operations applied to the state $\rho_{AB}$;

(3) $F^{(A)}(\rho_{AB})$ is monotonically decreasing under local quantum channels on subsystem $B$;

(4) $F^{(A)}(\rho_{AB})$ reduces to the so-called capacity of entanglement [41] for pure states $\rho_{AB}$.

We provide the proof in this appendix.

*Proof.* For the first part of (1), consider a classical-quantum state $\rho_{AB}$, we have $[\rho_{AB}, K_A] = 0$, which indicates that $\rho_{AB}$ is invariant under the modular flow. Thus $F^{(A)}(\rho_{AB}) = 0$ by Eq. (C1).

The second part of (1) follows from the property (1) of the IP. The IP must vanish because the QFI vanishes for a particular nondegenerate $\Gamma_A \sim \log \rho_A$.

(2) can be most easily seen from the basis independence of the QFI. More concretely, one can alternatively write the QFI as [48]

$$F_H = \lim_{dX \to 0} \frac{D_B(\rho(X - dX), \rho(X + dX))}{dX^2}, \tag{C1}$$

where $\rho(X) := e^{iK_A s} \rho_{AB} e^{-iK_A s}$. Such an expression is apparently invariant under a local change of basis,

(3) follows from the monotonicity of the QFI. For any Hamiltonian $H_A$ on $A$ and any state $\rho_{AB}$, $F_{H_A}(\rho_{AB})$ monotonically decreases under a quantum channel on $B$. This applies to $F^{(A)}(\rho_{AB})$, which in particular chooses $H_A = K_A$.

(4) follows from the fact that the QFI reduces the variance $F_H(\rho) = \text{Tr}(\rho H^2) - \text{Tr}(\rho H)^2$ for a pure state. If one chooses $H = K_A$, then the variance $\langle K_A^2 \rangle - \langle K_A \rangle^2$ is exactly the capacity of entanglement [41]. □