# Peer review of "Chirality, magic, and quantum correlations in multipartite quantum states"

_SciPost Physics_

## Round 1 · Referee Report · Beni Yoshida (Referee 1) · 2025-12-10

The referee discloses that the following generative AI tools have been used in the preparation of this report:
I used an AI tool (chatGPT) to correct minor typos and improve clarity in some part of my writing. All evaluations and judgments are entirely my own.
Strengths
-
Disclaimer: I am in the same institute as one of the authors. We are in different research groups and have never collaborated, though we do have occasional information interactions. I do not believe this affects my judgement of the work.
-
The paper introduces a number of novel and thought-provoking ideas, including 1) how to define "chirality" for a single wavefunction 2) notion of n-partite chirality 3) no-go theorem for qubit stabilizer state 4) relating chirality to magic 5) proposals/restrictions for potential chirality measures 6) there are chiral states which cannot be detected by modular commutator 7) and several additional insights throughout. In my opinion, some of these results will have long-standing impacts. Each of these ideas deserves to be further studied.
-
I particularly liked their characterization of chirality via local unitaries (finite-depth or n-partite). Also, the no-go for qubit stabilizer was a bit surprising (as 3F model is not chiral in this definition). In fact, these motivated me to extend some of their results by myself, which occupied quite a bit of my research time during the summer. I anticipate that several other directions suggested by this paper will be fruitful in the future.
-
Authors are true experts in this subject. They present physically insightful viewpoints while also providing rigorous/precise formulations when needed. The motivations behind each conceptual development are clearly spelled out.
Weaknesses
-
The primary weakness is that many of the key ideas are not fully developed. The paper presents a collection of interesting ideas (most of which are likely to become important with further study), but in their current form, each individual result feels somewhat preliminary. As a result, the central theme of the paper is not sharply defined. The paper would benefit from a more focused story, beginning with the title and abstract, and continuing through the main presentation.
-
Although the authors focus on resource theoretic aspects of chirality, these are, in my opinion, not the most compelling or novel aspects of the work. The foundational ideas concerning chirality and entanglement are conceptually deeper and more innovative.
Report
Requested changes
No change is needed.
Recommendation
Publish (surpasses expectations and criteria for this Journal; among top 10%)

---

## Editorial Decision

unknown